# Species and habitat specific changes in bird activity in an urban environment during Covid 19 lockdown

Congnan Sun[1,2,3], Yoel Hassin[1], Arjan Boonman[1], Assaf Shwartz[4], Yossi Yovel[1,5,6]*

[1]School of Zoology, Faculty of Life Sciences, Tel Aviv University, Tel Aviv, Israel; [2]College of Life Sciences, Hebei Normal University, Shijiazhuang, China; [3]Hebei Collaborative Innovation Center for Eco-Environment, Hebei Normal University, Shijiazhuang, China; [4]Faculty of Architecture and Town Planning, Technion, Israel Institute of Technology, Haifa, Israel; [5]The Steinhardt Museum of Natural History, National Research Center for Biodiversity Studies, Tel-Aviv University, Tel Aviv, Israel; [6]Sagol School of Neuroscience, Tel Aviv University, Tel Aviv, Israel

*For correspondence: yossiyovel@gmail.com

Competing interest: The authors declare that no competing interests exist.

**Abstract** Covid-19 lockdowns provided ecologists with a rare opportunity to examine how animals behave when humans are absent. Indeed many studies reported various effects of lockdowns on animal activity, especially in urban areas and other human-dominated habitats. We explored how Covid-19 lockdowns in Israel have influenced bird activity in an urban environment by using continuous acoustic recordings to monitor three common bird species that differ in their level of adaptation to the urban ecosystem: (1) the hooded crow, an urban exploiter, which depends heavily on anthropogenic resources; (2) the rose-ringed parakeet, an invasive alien species that has adapted to exploit human resources; and (3) the graceful prinia, an urban adapter, which is relatively shy of humans and can be found in urban habitats with shrubs and prairies. Acoustic recordings provided continuous monitoring of bird activity without an effect of the observer on the animal. We performed dense sampling of a 1.3 square km area in northern Tel-Aviv by placing 17 recorders for more than a month in different micro-habitats within this region including roads, residential areas and urban parks. We monitored both lockdown and no-lockdown periods. We portray a complex dynamic system where the activity of specific bird species depended on many environmental parameters and decreases or increases in a habitat-dependent manner during lockdown. Specifically, urban exploiter species decreased their activity in most urban habitats during lockdown, while human adapter species increased their activity during lockdown especially in parks where humans were absent. Our results also demonstrate the value of different habitats within urban environments for animal activity, specifically highlighting the importance of urban parks. These species- and habitat-specific changes in activity might explain the contradicting results reported by others who have not performed a habitat specific analysis.

## eLife assessment

This manuscript offers a **valuable** contribution to studying wildlife responses during and after COVID-19 lockdowns. It **convincingly** demonstrates that bird species in urban areas respond differently to human activity changes. What sets this study apart from others on avian responses to COVID-19 lockdowns is its use of passive acoustic monitoring. By concurrently measuring anthropogenic noise, a crucial reflection of changes in human activity due to COVID-19 lockdowns, this study reveals rare local-scale variations in bird responses to human activity. Only one study so far has used vocalization recordings to assess the effects of COVID-19 lockdowns on a bird species.

**eLife digest** Lockdowns due to the COVID-19 pandemic reduced human activity in early 2020, providing a rare opportunity to examine how wildlife behaves when humans are absent. While several studies reported increased abundance of animals in urban habitats, others cast doubt on these reports. Variation in study conclusions could be due to different species showing different levels of adaptation to human activity. Additionally, studies that rely on visually observing animals can impact their behavior and those based on public reporting may also have been influenced by changes in human activity. Therefore, it remained unclear whether COVID-19 lockdowns impacted wildlife and how this might differ among species.

To quantify wildlife presence and activity during lockdown, Sun et al. placed recording devices in different urban environments, including roads, residential areas, and urban parks across Tel Aviv in Israel during early 2020. This allowed continuous monitoring of bird vocalizations during lockdown and non-lockdown periods and ensured the birds were not disturbed by human observers.

Three common bird species, which each show different levels of adaptation to urban ecosystems, were monitored. The hooded crow, which depends heavily on human resources, and the rose-ringed parakeet, an invasive alien species which has adapted to exploit human resources, decreased their activity in most urban habitats during lockdowns. On the other hand, the graceful prinia, which has adapted to thrive in urban green spaces but is relatively shy of humans, showed increased activity, especially in parks where humans were absent.

The findings of Sun et al. reveal that birds show species- and habitat-specific changes to their behavior as a result of decreased human activity. This might explain why previous studies – which did not perform habitat-specific analyses – gave conflicting reports of the impact of COVID-19 lockdowns on wildlife activity. The results also demonstrate the value of different habitats within urban environments for animal activity, specifically identifying the importance of urban parks. By highlighting the impact of human activity on urban wildlife, the findings provide a scientific basis for future conservation efforts.

## Introduction

Covid 19 lockdowns provided a unique opportunity to examine the effects of humans on wildlife presence and activity (*Rutz et al., 2020*). The public media was full of reports on animals that have supposedly taken over areas that are most of the time occupied by human activity. Several studies reported animals expanding their behavior to day-time (*Manenti et al., 2020*), or venturing deeper into urban areas during the COVID-19 lockdowns (*Vardi et al., 2021*). Several studies reported an increase in avian abundances in urban habitats (*Bates et al., 2021*), while other studies casted doubts on these reports (*Vardi et al., 2021*; *Gordo et al., 2021*). These discrepancies in the responses of species to the lockdowns could stem from the fact that species may differ in their levels of adaptation to human activity (*Lowry et al., 2013*). Furthermore, most studies were performed in large scale with widely spread sampling points, and focused on specific urban habitats such as large green spaces which represent only a small fraction of the diverse urban landscape (*Shwartz et al., 2014*). There is therefore a need to fine-scale studies to explore how species with different levels of adaptation to human activity were affected by the lockdowns across habitats neighboring one-another. Such research can help advancing the understanding of the effect of humans on wildlife presence and activity. Moreover, most previous studies have relied on human (citizen) sporadic reports, which can be biased as they were also affected by changes in human activity (*Vardi et al., 2021*), or on observer-based surveys, which may affect bird activity (*Ross and Browning, 2017*). Using passive acoustic recordings, as we did, holds an interesting opportunity to tackle some of these issues and allows for more objective disturbance-free monitoring of wildlife activity (*Ross and Browning, 2017*).

Acoustic monitoring has been on the rise as a useful method for assessing insect (*Penone et al., 2013*), bat (*López-Bosch et al., 2022*), and bird (*Pérez-Granados et al., 2021*) diversity. This methodology has several clear advantages: it is continuous and does not require the observer to be present and thus the animals are not affected by the measurement itself (*Shonfield and Bayne, 2017*; *Gibb et al., 2019*). Acoustic recordings are cost-effective, especially with the development of modern cheap recorders, and they are efficient in dense environments where the animals cannot be easily detected

such as dense forests or urban environments. An additional advantage is that the data collected using acoustic recordings can be reanalyzed and reinterpreted as new questions arise, or when looking for new species (*Borker et al., 2014*; *Pérez-Granados and Schuchmann, 2020*). These recordings can also be used to monitor human activity, allowing to evaluate the impact of local human disturbance on species (*Buxton et al., 2018*). Finally, they allow estimating ambient noise levels (*Alfaro-Rojas et al., 2020*), as well as changes in the acoustics of the vocalizations, which can indicate adaptations of animal communication (*de Framond and Brumm, 2022*). Thus, acoustic recordings provide an excellent opportunity to monitor various species across urban habitat in a standardized manner.

In this study, we aimed to examine the effects of the Covid 19 lockdown on the activity of species that represent different levels of adaption to humans and exploitation of the urban resources across different types of urban habitats in Tel-Aviv, Israel. Research exploring biodiversity response to urbanization has classified species into three main groups regarding their level of adaptation to human activity and their ability to exploit anthropogenic resources (*Blair, 1996*; *Kark et al., 2007*): (1) urban avoiders, species that are particularly sensitive to human induced changes and reach their highest densities in natural environments outside the urban area; (2) urban adapters, species that thrive in urban green spaces, as they have adapted to exploit green urban resources (e.g., ornamental vegetation); and (3) urban exploiters, species that adapted to exploit the resources provided by humans in the urban environment and therefore reach high densities in the most built and developed environments. These species together with non-native invasive alien species that have also adapted to exploit various anthropogenic green and grey resources often replace a wider range of native species in a process that was describe as the biotic homogenization (*Lockwood and McKinney, 2001*; *Crooks et al., 2004*).

Building on previous studies that were conducted in Israel and classified common bird species to these three groups (*Kark et al., 2007*; *Colléony and Shwartz, 2020*), we selected three widespread and common urban species that vary in their level of adaptation to human activity. The hooded crow (*Corvus corone cornix*) is a native urban exploiter that heavily depends on anthropogenic resources for foraging and breeding and can be found in all types of urban habitats (*Kark et al., 2007*). The rose-ringed parakeet (*Psittacula krameri*) is a non-native urban adaptor that thrives in Israel and elsewhere

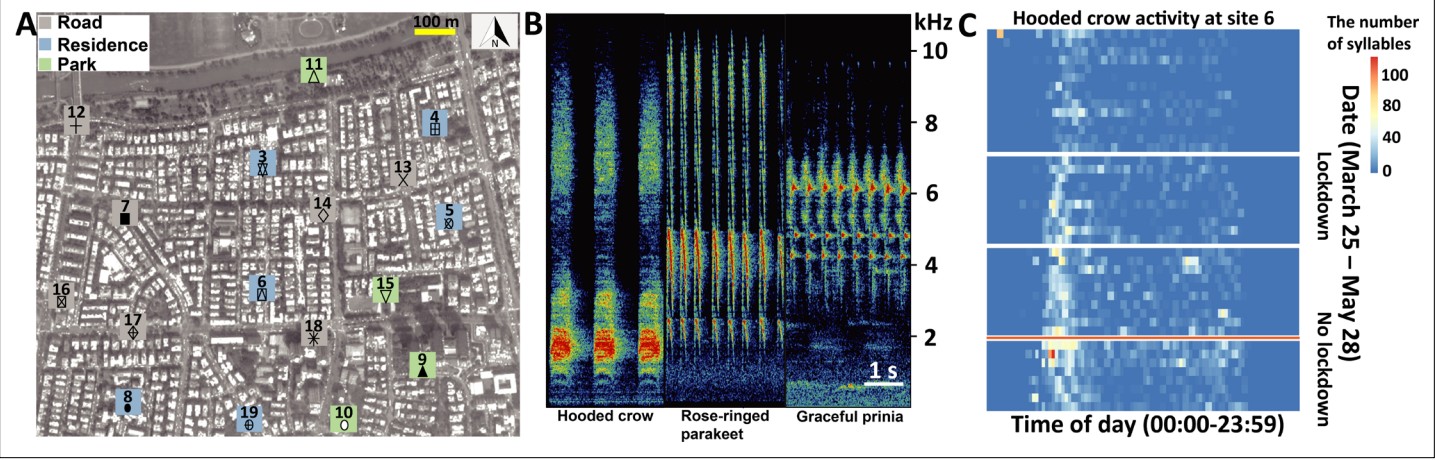

**Figure 1.** Study system, vocal characters, and vocal activities. (**A**) The seventeen sampling sites (Grey square: road; Blue square: residence; Green square: Park). The total activity for each species at each location is shown in bars for the lockdown (red) and no-lockdown (blue) periods. Different sampling sites are represented by different symbols. (**B**) Spectrograms of vocalizations with Hooded crow, Rose-ringed parakeet, and Graceful prinia vocalizations. (**C**) An example of a heat-map indicating the activity of Hooded crows at site 11 monitored continuously between March 31 and May 28 (without the dates between April 9 – April 23, May 4 – May 6, May 17 – May 20, depicted by white gaps). The x-axis depicts the time of day (not normalized to sunrise). The y-axis is the date. The red horizontal line separates lockdown from no lockdown periods. The figure also suggests an overall seasonal increase in activity, but our models suggest a lockdown effect on top of this seasonal effect.

The online version of this article includes the following source data for figure 1:

**Source data 1.** The activity of hooded crow at 17 sites in central Tel-Aviv.

**Source data 2.** The activity of rose-ringed parakeet at 17 sites in central Tel-Aviv.

**Source data 3.** The activity of graceful prinia at 17 sites in central Tel-Aviv.

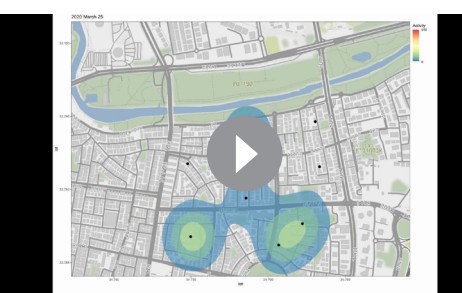

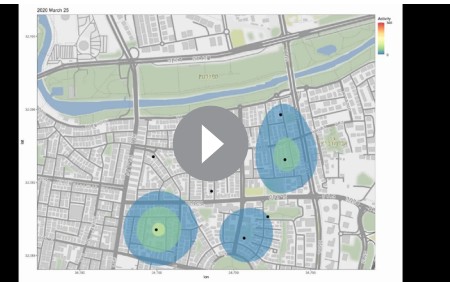

**Video 1.** Dynamic illustrations of the changes in crow's activity.
https://elifesciences.org/articles/88064/figures#video1

**Video 2.** Dynamic illustrations of the changes in parakeet's activity.
https://elifesciences.org/articles/88064/figures#video2

due to its ability to adapt and exploit different types of human resources (e.g. ornamental plants for feeding and infrastructure of breeding; *White et al., 2019*). The Graceful prinia (*Prinia gracilis*) is an insectivorous species that is relatively shy of humans and can be found in various green spaces where there are shrubs and prairies that represent its main habitat. The first two species can be considered as urban exploiters or synanthropic species that thrive in urban areas, and their population is increasing in Israel, while the third one represents an urban adaptor with a population that is currently decreasing in Tel-Aviv and across the country (*Colléony and Shwartz, 2020*). All birds were within their breeding season during both the lockdown and non-lockdown periods (Materials and methods).

We performed our study around the first lockdown period in Israel (March-May 2020). We selected common vocalizations emitted by each species usually in intra-specific contexts (see Materials and methods). To reduce the effect of ambient noise on our findings, we focused on loud vocalizations that were high above the noise level. The exact bird detection range of our method is hard to estimate accurately (because it depends on many parameters such as the building coverage) but we estimate it to be ~50 m. In light of the complex mosaic of urban habitats, we compared activity in three micro-habitats within cities, namely, in parks, roads and residential areas. We performed our monitoring at 17 sites all within the urban environment of central Tel-Aviv on average ~200 m apart from each other. Thus, our sampling was much denser than most previous studies allowing us to examine changes over fine spatial scales. The region we monitored included residential areas with small parks scattered within them and with small-medium sized roads connecting them. Buildings in this region are usually surrounded by small gardens, which will typically have some fruit and other trees. The north-most sector of the region includes a large open habitat on the bank of the Yarkon river (the Yarkon Park), recognized as an important bird resource in Tel-Aviv that serves as green corridor connecting the city to more natural environments around it (*Figure 1A*; *Shwartz et al., 2008*).

We quantified birds' presence and found changes in bird activity during the lockdown period that were species and habitat specific. We found that human following (exploiter) species probably moved their activity to residential areas during the lockdown, in contrast to a human aversive species (an adapter) that seemingly increased its activity in all habitats in the absence of humans. Other environmental factors, such as ambient temperature, also influenced bird activity, but the effect of the lockdown was central even after controlling for these variables. Our results thus demonstrate the importance of micro-habitats within the urban environment and highlight the complexity of nature's response to a dramatic shift in human activity.

## Results

We analyzed 388,080 audio recordings accounting for a total duration of 3234 hr from 17 sites (*Figure 1A*). In total we detected 52,080 files with 72,824 syllables of Hooded crows, 33,654 files with 73,445 syllables of Rose-ringed parakeets

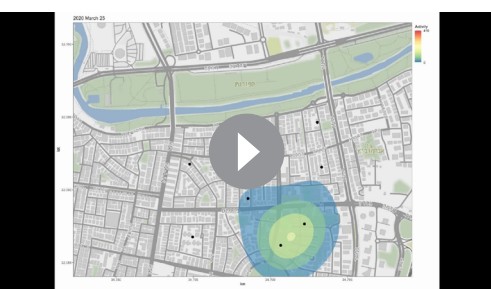

**Video 3.** Dynamic illustrations of the changes in prinia's activity.
https://elifesciences.org/articles/88064/figures#video3

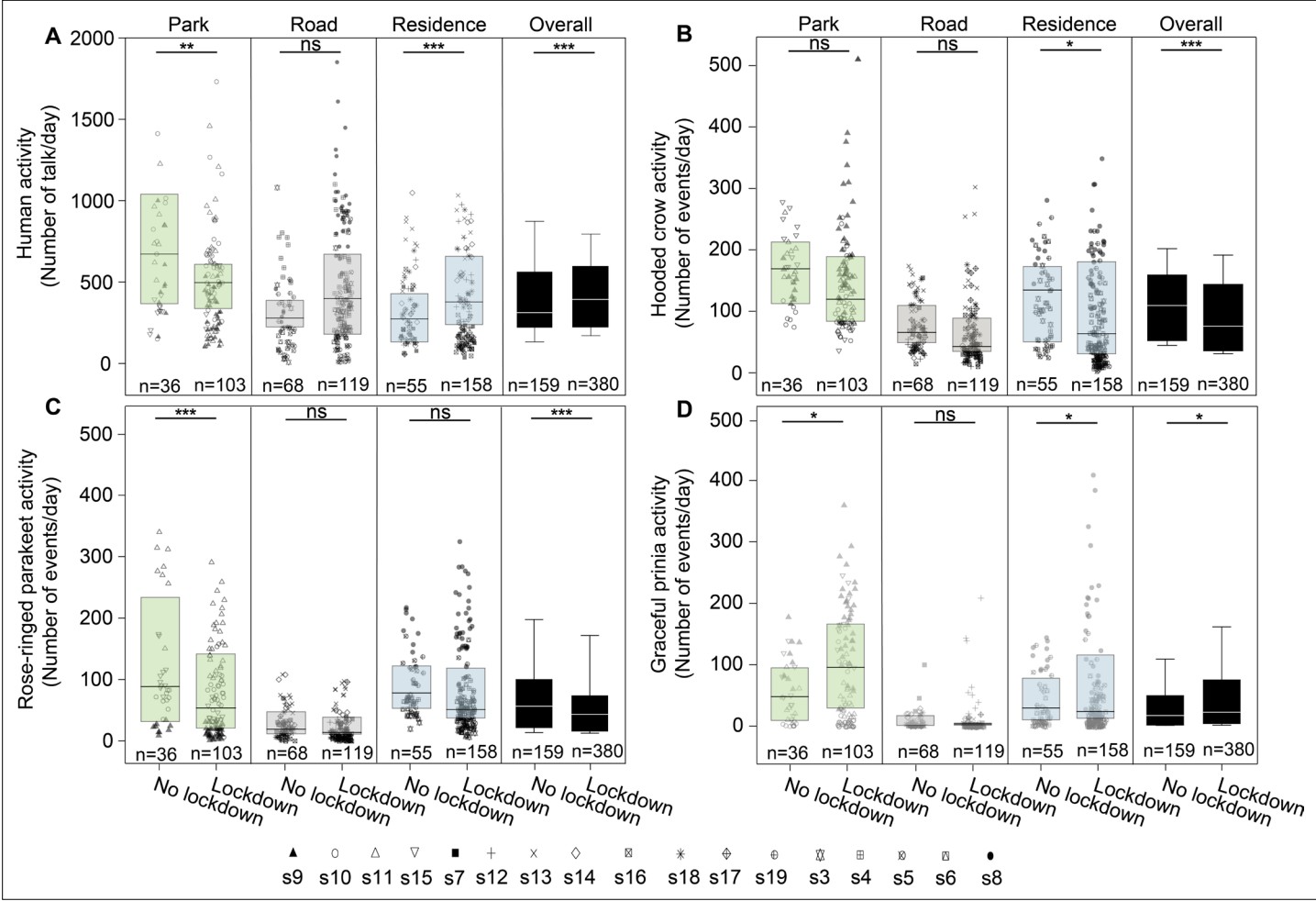

**Figure 2.** Activity of birds. Boxplots show the activity of (**A**) Human activity, (**B**) Hooded crow, (**C**) Rose-ringed parakeet, and (**D**) Graceful prinia for each site category during no lockdown and during lockdown. Green box plot: park; Grey box plot: road; Blue box plot: residence; Black box plot: overall. Note that human activity was assessed based on human speech so the increase observed during lockdowns in roads represents pedestrian and not car activity. Box plot lower and upper box boundaries show the 25th and 75th percentiles, respectively, with the median inside. The lower and upper error lines depict the 10th and 90th percentiles, respectively. Outliers of the data are shown as black dots. Different sampling sites represented by different symbols. Except for roads (GLMM, p=0.632), there were significant differences in human activity between no lockdown period and lockdown period for all sites (GLMM, p<0.0001), for parks (GLMM, p=0.002) and for residential sites (GLMM, residential sites: p<0.0001). In [B], [C] and [D] post hoc tests are indicated with significance levels: *p<0.05, **p<0.01, ***p<0.001, not significant [ns]. n indicates the number of sampling days.

The online version of this article includes the following figure supplement(s) for figure 2:

**Figure supplement 1.** Activity of birds.

**Figure supplement 2.** Activity of birds.

**Figure supplement 3.** Activity of birds.

**Figure supplement 4.** Activity of birds.

**Figure supplement 5.** Activity of birds.

**Figure supplement 6.** Activity of birds.

and 21,800 files with 117,982 syllables of Graceful prinias. The distribution of the audio recordings along the sampling periods can be found in **Supplementary file 1a-c**. A detailed summary of the activity of each species at each site can be found in **Figure 1—source data 1–3** (and in **Figure 1A**). We used these vocalizations to assess birds' spatio-temporal activity. We referred to the total number of events (i.e. files) detected per species as its total daily activity. We also estimated daily activity variability by calculating the coefficient of variance of number of daily events. Dynamic illustrations of the changes in bird activity are presented in **Videos 1–3**. In addition, human activity significantly increased

in residential areas by 49% during the lockdown and significantly decreased in parks by 31% as visiting parks was forbidden as expected (*Figure 2A*; *Supplementary file 1d*).

## Bird activity

When analyzing the activity of all species together, we found that the bird species and site type (park, road, residential) had a significant effect on activity, and moreover, that the interaction between the lockdown status and the species was also significant (*Table 1*). Below, we thus analyze the effect of the lockdown status on each species of bird separately. However, a few general patterns were observed from the analysis of all species together. Bird activity (number of daily events) was overall always highest in parks (62% more than near roads and 53% more than in residential areas, *Table 1*). Bird activity significantly decreased with an increase in ambient noise levels (*Table 1*). The average effect size was a reduction of 6% in activity per 1 dB of noise. As expected, ambient noise levels during the lockdown were significantly lower than during the non-lockdown period by an average of ca. 1.5 dB for crows, 1.7 dB for parakeets, and 1.7 dB for Prinia (LMM: –1.5<estimate < –1.3,–7.5<t < –5.2, p<0.0001; the noise was estimated for each species in its call's peak frequency, *Figure 3— figure supplement 1*), but this reduction in noise was not enough to explain changes in bird activity during the lockdown (*Table 1*). The overall bird activity also significantly increased with an increase in temperature (by 5% per degree, *Table 1*).

We also examined the variability of the activity for all species together. Note that a reduction in the coefficient of variance per day means that activity is more constant, or less variable during the day. Overall, the coefficient of variance showed opposite patterns in comparison with the activity, that is, whenever we observed more activity, we observed less variability and vice versa. When analyzing all species together, we found that the activity in roads and residential areas was significantly more variable than in parks (*Table 1* and *Supplementary file 1e*). Moreover, the variability significantly increased with an increase in noise, it increased during the lockdown periods, and it significantly decreased with an increase in temperature (*Table 1* and *Supplementary file 1e*). As with the total activity, the patterns were species and site specific, as analyzed and detailed below.

Given the differences between species and site type, we analyzed the effect of the above-mentioned parameters on activity and activity variability of each species. Hooded crow activity significantly decreased (by ~48%) during the lockdown period (*Figure 2B* and *Figure 1C*), but the effect of the lockdown was not apparent at all site types (*Table 1* for the full list of model parameters, and *Supplementary file 1e* for the model selection results). Post-hoc analysis revealed that while hooded crow activity decreased at all site types during the lockdown, the decrease was only significant in residential areas (by ~39%), and marginally significant in parks (*Figure 2B*; see *Supplementary file 1f* for full details). Hooded crows seemed to get used to the lockdown as activity increased along the lockdown period (GLMM: lockdown-countdown interaction, p<0.001; *Table 1*). Note that our statistical models already take many of the lockdown ambient parameters into account including noise and human activity, so the significance we observe for the lockdown status might be due to additional lockdown-related parameters (see Discussion). In terms of their variability of activity, crows generally followed the opposite patterns with the variability behaving oppositely to the activity, and showing more variability with an increase in noise levels (*Table 1* and *Supplementary file 1e*).

Similar to Hooded crows, the activity of rose-ringed parakeets decreased during the lockdown in all site types, but this decrease was only significant in parks (by ~90%, see *Figure 2C* and *Supplementary file 1f*). Parakeet activity also significantly increased with an increase in temperature (*Table 1*, *Supplementary file 1f*). In terms of variability of activity, parakeets exhibited a significant decrease in variability during lockdown periods in residential areas and in parks, and no significant change near roads (*Supplementary file 1f*). Parakeets exhibited more activity variability with an increase in ambient noise (*Table 1*, *Supplementary file 1e*) and exhibited less variability with an increase in temperature (*Table 1*, *Supplementary file 1e*).

Graceful prinias showed opposite patterns from both previous species exhibiting a significant increase in activity during the lockdown periods when examining all sites together (*Table 1*). When examining the site types separately, Prinia activity increased significantly during the lockdown in parks (by ~12%) and the increase in prinia activity in residential areas was marginally significantly (by ~7%), while there was no change near roads (*Figure 2D*; *Supplementary file 1f*). Prinia activity decreased significantly with an increase in noise levels (*Table 1*) and significantly increased with an increase in

**Table 1.** Effects of predictor variables on birds' activity based on generalized and general linear mixed models (GLMM and LMM). Estimates were calculated in % per day for the following units: Temperature – per degree, Noise – per dB, Human activity – per 1 talking event, Lockdown related parameter – per existence of the lockdown (yes/no).

| Species | Dependent variable | Predictors | Estimate | z | p | 95% CI | Percent |
|---|---|---|---|---|---|---|---|
| All species | **Activity-** Number of events/day | (Intercept) | 7.850 | 22.541 | <0.001 | - | - |
| | | Bird_species | –0.623 | –84.742 | **<0.001** | - | 46.367 |
| | | Lockdown_status | –0.404 | –2.623 | **0.009** | - | 33.236 |
| | | Human_activity | –0.00001 | –0.539 | 0.590 | - | 0.001 |
| | | Noise | –0.062 | –20.625 | **<0.001** | - | 6.012 |
| | | Temperature | 0.045 | 4.194 | **<0.001** | - | 4.649 |
| | | Site_category_residence | –0.739 | –2.764 | **0.006** | - | 53.286 |
| | | Site_category_road | –0.923 | –3.558 | **<0.001** | - | 62.075 |
| | | Lockdown_status Count_down* | 0.008 | 3.820 | **<0.001** | - | 0.835 |
| | | Lockdown_status Site_category_residence* | –0.072 | –3.435 | **0.001** | - | 7.294 |
| | | Lockdown_status Site_category_road* | 0.094 | 4.074 | **<0.001** | - | 10.447 |
| | | Lockdown_status Noise* | –0.007 | –2.723 | **0.006** | - | 0.746 |
| | | Lockdown_status Human_activity* | 0.0003 | 14.074 | **<0.001** | - | 0.032 |
| | | Bird_species Lockdown_status* | 0.253 | 29.155 | **<0.001** | - | 31.379 |
| | †**Activity variability-** CV of the number of events/day | (Intercept) | –0.963 | 5.975 | <0.001 | −1.282,−0.618 | - |
| | | Bird_species | 0.208 | 16.877 | **<0.001** | **0.184, 0.233** | - |
| | | Lockdown_status | 0.565 | 4.238 | **<0.001** | **0.269, 0.839** | - |
| | | Noise | 0.025 | 8.554 | **<0.001** | **0.0198, 0.030** | - |
| | | Site_category_residence | 0.170 | 2.403 | **0.016** | **0.030, 0.310** | - |
| | | Site_category_road | 0.224 | 3.238 | **0.001** | **0.089, 0.361** | - |
| | | Human_activity | –0.00004 | 1.150 | 0.250 | –0.0001, 0.00003 | - |
| | | Temperature | –0.010 | 3.394 | **0.001** | **−0.016,−0.004** | - |
| | | Bird_species Lockdown_status* | –0.057 | 3.888 | **<0.001** | **−0.086,−0.028** | - |
| | | Count_down Lockdown_status* | –0.001 | 1.503 | 0.133 | –0.003, 0.0004 | - |
| | | Lockdown_status Noise* | –0.008 | 3.157 | **0.002** | **−0.013,−0.003** | - |
| | | Lockdown_status Human_activity* | –0.00005 | 1.308 | 0.191 | –0.00018, 0.00003 | - |
| | | Lockdown_status Site_category_residence* | 0.023 | 0.641 | 0.522 | –0.048, 0.098 | - |
| | | Lockdown_status Site_category_road* | 0.036 | 0.995 | 0.320 | –0.042, 0.106 | - |

*Table 1 continued*

| Species | Dependent variable | Predictors | Estimate | z | p | 95% CI | Percent |
|---|---|---|---|---|---|---|---|
| Hooded crow | †**Activity-** Number of events/day | (Intercept) | 7.475 | 20.026 | <0.001 | 6.743, 8.206 | - |
| | | Lockdown_status | −0.647 | 4.508 | **<0.001** | **−0.928,−0.366** | 47.639 |
| | | Noise | −0.045 | 12.324 | **<0.001** | **−0.052,−0.038** | 4.400 |
| | | Site_category_residence | −0.790 | 2.374 | **0.018** | **−1.443,−0.1386** | 54.616 |
| | | Site_category_road | −0.738 | 2.284 | **0.022** | **−1.372,−0.105** | 52.193 |
| | | Human_activity | −0.0001 | 3.376 | **0.001** | **−0.0002,−0.00004** | 0.010 |
| | | Count_down Lockdown_status* | 0.017 | 6.542 | **<0.001** | **0.012, 0.022** | 1.749 |
| | | Lockdown_status Site_category_residence* | −0.064 | 2.288 | **0.022** | **−0.119,−0.009** | 6.385 |
| | | Lockdown_status Site_category_road* | 0.221 | 7.606 | **<0.001** | **0.164, 0.278** | 25.722 |
| | | Lockdown_status Human_activity* | 0.0004 | 10.456 | **<0.001** | **0.0003, 0.0004** | 0.042 |
| | | Lockdown_status Noise* | −0.002 | 0.682 | 0.495 | −0.009, 0.0056 | 0.212 |
| | | Temperature | −0.009 | 0.645 | 0.519 | −0.037, 0.019 | 0.959 |
| | †**Activity variability-** CV of the number of events/day | (Intercept) | −1.447 | 7.373 | <0.001 | −1.827,−1.023 | - |
| | | Lockdown_status | 0.889 | 5.473 | **<0.001** | **0.534, 1.217** | - |
| | | Noise | 0.033 | 9.604 | **<0.001** | **0.026, 0.040** | - |
| | | Site_category_residence | 0.293 | 2.958 | **0.003** | **0.095, 0.486** | - |
| | | Site_category_road | 0.236 | 2.479 | **0.013** | **0.048, 0.423** | - |
| | | Human_activity | 0.00004 | 0.999 | 0.318 | −0.00005, 0.0001 | - |
| | | Lockdown_status Noise* | −0.015 | 4.618 | **<0.001** | **−0.021,−0.008** | - |
| | | Lockdown_status Human_activity | −0.0001 | 2.959 | **0.003** | **−0.0002,−0.00005** | - |
| | | Temperature | −0.003 | 1.074 | 0.283 | −0.010, 0.003 | - |
| | | Count_down Lockdown_status* | −0.001 | 1.032 | 0.302 | −0.003, 0.001 | - |
| | | Lockdown_status Site_category_residence* | 0.045 | 1.000 | 0.317 | −0.0414, 0.141 | - |
| | | Lockdown_status Site_category_road* | −0.009 | 0.204 | 0.838 | −0.095, 0.082 | - |

*Table 1 continued on next page*

*Table 1 continued*

| Species | Dependent variable | Predictors | Estimate | z | p | 95% CI | Percent |
|---|---|---|---|---|---|---|---|
| Rose-ringed parakeet | †**Activity-** Number of events/day | (Intercept) | 6.320 | 11.848 | <0.001 | 5.272, 7.373 | - |
| | | Lockdown_status | –0.106 | 0.379 | 0.704 | –0.656, 0.443 | 10.058 |
| | | Noise | –0.061 | 10.900 | **<0.001** | **–0.072,–0.050** | 5.918 |
| | | Site_category_residence | –0.493 | 1.175 | 0.240 | –1.314, 0.329 | 38.921 |
| | | Site_category_road | –1.201 | 2.949 | **0.003** | **–1.999,–0.403** | 69.911 |
| | | Human_activity | –0.0003 | 6.891 | **<0.001** | **–0.0004,–0.0002** | 0.030 |
| | | Temperature | 0.060 | 3.990 | **<0.001** | **0.031, 0.089** | 6.307 |
| | | Lockdown_status Noise | –0.016 | 3.268 | **0.001** | **–0.026,–0.006** | 1.635 |
| | | Lockdown_status Site_category_residence | 0.408 | 9.968 | **<0.001** | **0.328, 0.488** | 52.396 |
| | | Lockdown_status Site_category_road | 0.574 | 13.488 | **<0.001** | **0.491, 0.658** | 81.412 |
| | | Lockdown_status Human_activity | 0.001 | 15.304 | **<0.001** | **0.0006, 0.0008** | 0.106 |
| | | Count_down Lockdown_status | 0.004 | 1.234 | 0.217 | –0.002, 0.011 | 0.429 |
| | †**Activity variability-** CV of the number of events/day | (Intercept) | –0.460 | 3.225 | 0.001 | –0.751,–0.178 | - |
| | | Lockdown_status | 0.249 | 2.775 | **0.006** | **0.065, 0.442** | - |
| | | Noise | 0.020 | 8.303 | **<0.001** | **0.015, 0.025** | - |
| | | Site_category_residence | 0.090 | 1.054 | 0.292 | –0.078, 0.256 | - |
| | | Site_category_road | 0.196 | 2.376 | **0.018** | **0.032, 0.357** | - |
| | | Human_activity | 0.00001 | 0.419 | 0.675 | –0.00005, 0.00007 | - |
| | | Temperature | –0.007 | 2.331 | **0.020** | **–0.012,–0.001** | - |
| | | Count_down Lockdown_status* | –0.003 | 4.333 | **<0.001** | **–0.005,–0.002** | - |
| | | Lockdown_status Site_category_residence* | –0.051 | 1.818 | 0.069 | –0.105, 0.005 | - |
| | | Lockdown_status Site_category_road* | –0.109 | 3.915 | **<0.001** | **–0.164,–0.053** | - |
| | | Lockdown_status Human_activity* | –0.0001 | 3.108 | **0.002** | **–0.0002,–0.00004** | - |
| | | Lockdown_status Noise* | –0.002 | 0.866 | 0.387 | –0.008, 0.0033 | - |

*Table 1 continued*

| Species | Dependent variable | Predictors | Estimate | z | p | 95% CI | Percent |
|---|---|---|---|---|---|---|---|
| Graceful prinia | **Activity**- Number of events/day | (Intercept) | 3.314 | 2.245 | 0.025 | - | - |
| | | Lockdown_status | –0.352 | –0.595 | 0.552 | - | 29.672 |
| | | Human_activity | 0.00017 | 2.367 | **0.018** | - | 0.017 |
| | | Noise | –0.165 | –17.114 | **<0.001** | - | 15.211 |
| | | Temperature | 0.365 | 6.533 | **<0.001** | - | 44.051 |
| | | Site_category_residence | –1.012 | –1.440 | 0.150 | - | 64.287 |
| | | Site_category_road | –1.484 | –2.164 | **0.030** | - | 78.874 |
| | | Lockdown_status Count_down* | –0.104 | –14.979 | **<0.001** | - | 10.174 |
| | | Lockdown_status Site_category_residence* | –0.062 | –1.159 | 0.247 | - | 6.252 |
| | | Lockdown_status Site_category_road* | –1.203 | –13.290 | **<0.001** | - | 73.469 |
| | | Lockdown_status Noise* | 0.066 | 7.072 | **<0.001** | - | 7.232 |
| | | Lockdown_status Human_activity* | 0.003 | 3.513 | **<0.001** | - | 0.300 |
| | †**Activity variability**- CV of the number of events/day | (Intercept) | 0.920 | 1.496 | 0.135 | –0.284, 2.100 | - |
| | | Lockdown_status | –0.194 | 0.655 | 0.513 | –0.782, 0.425 | - |
| | | Noise | 0.018 | 1.823 | 0.068 | –0.002, 0.038 | - |
| | | Site_category_residence | 0.210 | 0.946 | 0.344 | –0.227, 0.643 | - |
| | | Site_category_road | 0.556 | 2.452 | **0.014** | **0.117, 1.009** | - |
| | | Temperature | –0.031 | 2.592 | **0.010** | **−0.054,–0.007** | - |
| | | Count_down Lockdown_status* | 0.005 | 1.578 | 0.115 | –0.001, 0.011 | - |
| | | Lockdown_status Site_category_residence* | 0.172 | 1.461 | 0.144 | –0.058, 0.406 | - |
| | | Lockdown_status Site_category_road* | 0.274 | 2.089 | **0.037** | **0.0163, 0.532** | - |
| | | Lockdown_status Noise* | –0.005 | 0.403 | 0.687 | –0.029, 0.020 | - |
| | | Human_activity | –0.00004 | 0.437 | 0.662 | –0.0003, 0.0002 | - |
| | | Lockdown_status Human_activity* | 0.00007 | 0.394 | 0.694 | –0.0003, 0.0004 | - |

CV: coefficient of variance.

*Interaction effect. 95% confidence intervals of the parameters that did not overlap zero are indicated in bold.

†model average.

temperature (*Table 1*). The increase during lockdown (vs. no lockdown) could not be explained as a result of seasonality because overall, prinia activity increased over time and the no-lockdown period was later in the season than the lockdown period. Prinia activity variability did not change significantly during lockdown periods (LMM: p=0.552; *Table 1*). Prinia showed less variability with an increase in temperature (LMM: p<0.001; *Table 1*). When examining each site type separately, activity variability did not change in residential sites and in roads but significantly decreased in parks (*Supplementary file 1f*).

We ran two models to control for the effect of our uneven sampling of the lockdown and no-lockdown periods: (1) We randomly selected five 10 day periods within the lockdown period and compared them to the no-lockdown period (where we also sample 10 days). This analysis revealed very similar results to the ones described above (for all five sub-samples) suggesting that our results were not an artifact of the imbalanced sampling (*Figure 2—figure supplements 1–5*); This approach was chosen over taking the last 10 days of the lockdown period, because towards the end of the lockdown, its enforcement was loosened and people gradually returned to their normal (pre-lockdown) behavior. (2)

**Table 2.** Effects of predictor variables on the root mean square (RMS) and peak frequency in three bird species based on the linear mixed models (LMM).

| Species | Dependent variable | Predictors | Estimate | t | p |
|---|---|---|---|---|---|
| Hooded crow | RMS | Lockdown_status | 0.064 | 0.148 | 0.883 |
| | | Noise | 0.354 | 7.054 | **<0.0001** |
| | Peak frequency | Lockdown_status | –5.99E-04 | –0.601 | 0.552 |
| | | Noise | 9.17E-05 | 0.728 | 0.468 |
| Rose-ringed parakeet | RMS | Lockdown_status | 1.065 | 3.290 | 0.001 |
| | | Noise | 0.470 | 7.772 | **<0.0001** |
| | Peak frequency | Lockdown_status | 3.93E-03 | 3.235 | **0.001** |
| | | Noise | 1.34E-04 | 0.783 | 0.435 |
| Graceful prinia | RMS | Lockdown_status | 0.208 | 0.475 | 0.635 |
| | | Noise | 0.844 | 9.449 | **<0.0001** |
| | Peak frequency | Lockdown_status | 1.31E-04 | 0.083 | 0.935 |
| | | Noise | 2.40E-03 | 0.928 | 0.357 |

we performed a permutation test where we permuted the lockdown status of each day (while maintaining the number of lockdown and no-lockdown days constant). We performed 1000 permutations and ran the GLMM model for each of the permuted data-sets (the GLMM was identical to the one presented in *Table 1*). We found that the estimate of the effect of lockdown was highest for the original data – higher than in all 1000 permutations, suggesting that there was an additional effect of the lockdown on top of potential other effects such as the changes in noise and temperature.

To assess the relative importance of the various parameters which contributed to the best explanatory model, we used the parameters of the best model (site category, ambient noise, human activity, temperature and lockdown status) and ran a discriminant function analysis (DFA) aiming to classify the level of bird activity (see Materials and methods). The DFA's managed to significantly classify bird activity for all three species 62.7% for Hooded crows, 80.7% for Rose-ringed parakeets and 82.5% for Gracefull prinias (binomial test: p<0.001 and see *Supplementary file 1g* for the full details). Analyzing the result showed that the first discriminant function explained a great majority of the variance and that it was affected (almost equally) by all for five parameters in a species-specific manner (see *Supplementary file 1g*).

## Bird acoustics

In light of several studies that have reported changes in bird vocal acoustics during COVID-19 lockdowns, we also tested this for the three species we studied, examining call intensity and peak frequency (the most intense frequency) for the recorded syllables (*Table 2*). All species seemed to produce louder vocalizations as ambient noise level increased. However, because they increased call intensity to a lesser degree than the additional noise, we suggest that this apparent increase was probably an artifact of the additional noise in the recordings rather than a real increase in source level. Specifically, we measured an increase of 0.35 dB/noise dB for crows, 0.47 dB/noise dB for parakeets, and 0.84 dB/noise dB for prinias (*Figure 3*, note that all slopes are ≤ 1; LMM: all p<0.0001 for all species; *Table 2*). We did not find any significant change in call frequency due to the lockdown (the vocalizations of rose-ringed parakeet had significantly lower peak frequencies during lockdown, but the difference was negligible: ~4 Hz, *Table 2*).

## Discussion

Animal activity during Covid-19 lockdowns has drawn much public attention with a popular notion that many species have reclaimed cities during lockdowns (*Díaz and Møller, 2023*; *Warrington et al., 2022*; *Behera et al., 2022*). Several reports however demonstrated that these claims may have

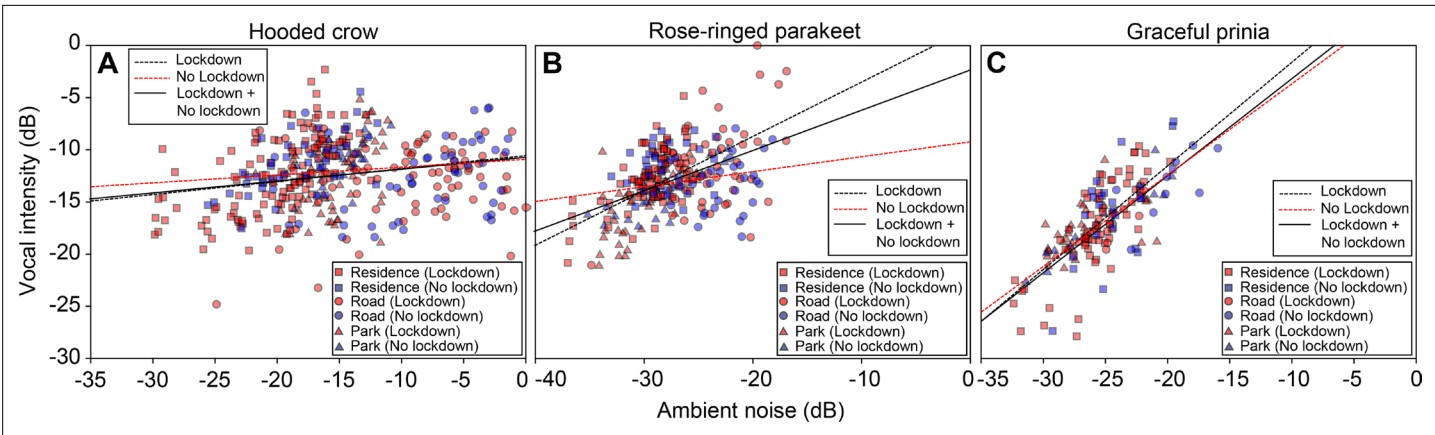

**Figure 3.** Vocal intensity as a function of ambient noise (normalized to maximum) for (**A**) Hooded crows. (**B**) Rose-ringed parakeets and (**C**) Graceful prinia. Red colors and blue colors represent data collected during no lockdown and during lockdown, respectively (each point represents the average over 1 day). Square: Residences; Circle: Roads; Triangle: Parks. The lines represent regression lines (solid line: Lockdown and No lockdown together; black dashed line: Lockdown; red black dashed line: No lockdown). The equations of the linear fits: y=0.124 x – 10.554 (Hooded crow; Lockdown; n=264); y=0.075 x – 10.877 (Hooded crow; No lockdown; N=119); y=0.521 x+1.705 (Rose-ringed parakeet; Lockdown; n=184); y=0.143 x – 9.204 (Rose-ringed parakeet; No lockdown; n=82); y=0.993 x+8.354 (Graceful prinia; Lockdown; n=108); y=0.876 x+5.110 (Graceful prinia; No lockdown; n=49). There were significant and positive relationships between ambient noise levels and vocal intensity for all bird species (LMM, Hooded crows: n=448; t=7.054, p<0.0001; Rose-ringed parakeets: n=266; t=7.772, p<0.0001; Graceful prinia: n=157; t=9.449, p<0.0001).

The online version of this article includes the following figure supplement(s) for figure 3:

**Figure supplement 1.** The ambient noise level (dB) between no lockdown period and lockdown period.

**Figure supplement 2.** The examples of different types of noise.

been exaggerated, and that species different responded differently to the sudden changes in human activity (*Vardi et al., 2021*). Yet, fine empirical evidence on how species activity changed during lockdown is scarce and knowledge is important to further understand how wildlife respond to human activity (*Montgomery et al., 2021*). Our fine-scale continuous acoustic monitoring of bird activity during lockdown and no-lockdown periods in various urban environments reveals a complex picture. On the one hand, synanthropic urban exploiter bird species such as the crows and parakeets, reduced their activity at all urban habitats. On the other hand, a non-synanthropic urban adapter songbird, which typically sings from within vegetation, the Graceful Prinia, showed almost opposite patterns, increasing its singing activity during lockdown both in parks and in residential areas. Our main conclusion is thus that responses vary between species with different levels of adaptation to humans but that they might depend on the specific habitat and also rely on many additional ambient parameters.

As expected, bird activity was correlated with several ambient factors such as temperature and noise (independently of the lockdown state). However, the changes in activity which we report for the lockdown period seem to be additional to changes in noise or temperature as these were accounted for in the models (but note that we used linear models and thus might have not fully accounted for them). Human presence (assessed by detecting human speech) could also not fully explain the shifts we observed in bird activity. Indeed, the activity of all species significantly correlated with human presence, but the presence of humans did not fully explain bird activity shifts. For instance, Crows and Parakeets decreased their activity in both parks and residential areas, even though human activity decreased in the former and increased in the latter. It is likely that additional factors that were altered by the lockdown such as the specific type of human activity (e.g. jogging and littering vs. walking) affected the birds' decision of activity. We also note that although we defined the lockdown as a binary state based on the official restrictions imposed by the government, in reality, humans gradually ignored parts of the restrictions in a non-binary manner. Thus, the last days of the lockdown period might have been more similar to the no-lockdown periods, weakening our results.

We used bird vocalizations as a proxy for activity. This method has been validated by many other recent studies where bird vocal activity was found to be positively correlated with the abundance of birds (see a Review in *Pérez-Granados and Traba, 2021* and also *Ducrettet et al., 2020*; *Pérez-Granados et al., 2021*; *Szymański et al., 2021*). Using vocalizations to estimate activity is

advantageous for many reasons as it allows collecting vast data over large spatio-temporal scales, but it also has its limitations. One challenge is distinguishing between changes in bird presence and changes in vocalizing, for example birds might be more or less vocal due to human activity without changing their position. We tried to rely on vocalizations that are used for intra-specific communication, but we could not assure that these vocalizations were not sometimes emitted towards predators such as humans. For parakeets, the great majority of vocalizations are used for communication between conspecifics (mostly in flight) and this is thus a good approximation for their presence and activity (*Pruett-Jones, 2021*). For crows, some of the vocalizations might be uttered during interactions with humans and thus, our results might be partially affected by the fact that there were less humans around during lockdown. Still, we argue that most crow vocalizations are directed towards conspecifics (*Palestrini and Rolando, 1996*) and thus acoustic monitoring is a good way to assess their presence and activity. Moreover, crow activity decreased in both parks and residential areas even though human activity showed opposite patterns in these sites. The vocalizations of prinia were territorial male songs, typically uttered towards conspecifics, so the increase we observed during lockdown, strongly suggests an increase in activity, even when accounting for the potential seasonal increase by adding temperature to the models. Bird activity is thus probably correlated with bird abundance, but we cannot determine for sure whether bird numbers changed during lockdown or whether the birds only changed their activity (i.e., interacted more and thus called more). Naturally, further studies are required to accurately connect changes in vocalization with changes in presence and activity. In addition to our analysis of the lockdown's effect, our results also clearly demonstrate the advantage of parks as safe havens for birds inside the urban environment – bird activity was always highest in parks regardless of lockdown condition (see *Table 1*). Interestingly, residential areas which in the Tel-Aviv area are characterized by many trees also revealed much bird activity, in some sites – not significantly less than in parks.

Unlike some previous studies (e.g. *Slabbekoorn and Peet, 2003*; *Brumm, 2004*; *Derryberry et al., 2020*), we did not find a significant change in bird vocal acoustics during the lockdown periods. Specifically, we observed an increase in vocalization amplitudes that was less than the increase in ambient noise. An increase in noise (at the same frequency band as the vocalization) will (by definition) increase the measured amplitudes of the vocalization making it impossible to detect a real increase in vocal intensity, unless the increase is larger than the increase in noise. Birds thus might have increased vocal intensity (minutely), but this could not have been detected using our method. Other studies that controlled for the recording distance (e.g. *Derryberry et al., 2020*) did find a change in intensity but this should not have affected our results regarding the frequency.

Various decisions must be made when using acoustic monitoring, such as when two adjacent recordings represent different individuals. We considered any recording of one of the three focal species within a file as one event thus ignoring the number of vocalizations within the file, however, we also tested a model which used the number of syllables (and not the number of files) as the explained variable, and this did not change the essence of the results (see *Figure 2—figure supplement 6* and *Supplementary file 1h-j*). The distribution of the birds' syllables along the sampling periods can be found in *Supplementary file 1k-m*. Notably, when using visual surveying, this problem also exists because usually individuals cannot be distinguished and thus an individual bird appearing twice in the images (or binoculars) might be counted as two birds. Another potential concern is that changes in ambient noise might influence the detection rates and thus the results. Increased noise levels might make it difficult to detect weak vocalizations and as a result imply a wrong reduction in activity. Because we used vocalizations that are far above the noise level of our recording system (see for example *Figure 1B*) the small changes observed in ambient noise could not explain our results. This also reduced the risk that our results were caused by a change in bird distribution in the area around the microphone (e.g. the birds were close or farther from the microphones). Moreover, if our findings were a result of birds escaping the presence of humans, we would have expected to see a large effect of human activity in the models, which was not the case. Finally, for some bird species, acoustic monitoring could also allow to distinguish between various behaviors, but we did not apply this approach here. Importantly, we find both an increase and a decrease in activity (depending on site type and species) so our results cannot be explained by the unidirectional change in noise (i.e. the decrease in noise during lockdown).

Another potential bias in our experiment was seasonality – the lockdown stretched over almost two months during which spring shifted to summer and temperature increased by an average of three degrees. To control for this, we added temperature (which is a proxy of seasonality) to the models. We also tested a model where the 5 days with extremely high temperatures were removed but the results did not change (see *Supplementary file 1n and o*). We also included the day since the beginning of the period (lockdown or not) to the model which could also account for seasonal effects. Moreover, in many cases, we found a decrease in activity during the later no-lockdown period, excluding the possibility that all our results could be explained by a seasonal effect.

Our results picture a complex and dynamic urban ecological system where animal activity depends on species and site, as well as on environmental parameters and heterospecific (e.g. human) activity. We moreover demonstrate that bird activity patterns can differ on a local scale only hundreds of meters apart. This complexity might explain the contradicting effects of lockdown reported by others (*Gordo et al., 2021*; *Estela et al., 2021*; *Bates et al., 2021*; *Manenti et al., 2020*). It points towards the importance of monitoring activity in fine-scale across geographical and environmental landscapes and of including environmental parameters in statistical models. Accurate and automated long-term methods to estimate biodiversity done in parallel to environmental measurements are going to be essential for monitoring future effects of global changes.

## Materials and methods
### Study site, species, and period
The study was conducted at 17 different sites in Tel-Aviv, Israel (*Figure 1A* and *Figure 1—source data 1–3*). The sites represent three different common habitats in this area: small parks, residential areas, and roads. They were randomly selected so that they are evenly distributed within the region of the study. We sampled four parks (sites 9, 10, 11, 15), seven roads (sites 7, 12, 13, 14, 16, 17, 18) and six residential sites (sites 3, 4, 5, 6, 8, 19). A device placed in a fifth park was stolen. We focused on three bird species that are common in this urban environment, that is Hooded crow, Rose-ringed parakeet and Graceful prinia. The vocalizations of these species are loud, highly repeatable and stereotypic (*Figure 1B*), making them rather easy for automatic recognition based on template matching (see next section).

Spreading over a 2-month period, the first lockdown in which we performed our recordings had different levels of severity. In the beginning, the restrictions were strictly enforced with traffic on the roads decreasing almost to zero and with severe limitations imposed on human activity including a curfew preventing people from moving more than 100 m from home. Use of the parks was also not allowed and accordingly, they were empty. Over time though, the enforcement was loosened, and human activity gradually returned to normal until the official removal of the lockdown. Using acoustic monitoring and automatic identification of three bird species, we compared bird and human activity during lockdown to a short period immediately after lockdown assuming that changes in activity between these adjacent time periods might be attributed to the lockdown (rather than to other environmental factors). We also account for seasonality by including environmental parameters (e.g. ambient temperature) in our models. It is important to note that all recordings were made during the breeding season of the three species. We would like to note that in terms of breeding, all birds were within the same state during both the lockdown and the non-lockdown periods. Parakeets and crows have a long breeding season Feb-end of June with one cycle. They will stay around the nest throughout this season and especially in the peak of the season March-May. Prinias start slightly later at the beginning of March with 2–3 cycles till end of June.

### Acoustic survey
At each site, we installed one recording device, Audiomoth (*Hill et al., 2018*), on a tree trunk or a hedge at 2–4 m above the ground. We set the recorders to record 30 s every 2 min (30 s of recording, 1.5 min of pause) continuously during day and night with a sampling frequency of 192 kHz. Recording was performed in 4 time-bouts during spring 2020, the peak of birds breeding season. Three of the sampling periods were during the lockdown: 25.3.2020–9.4.2020 (15 days), 24.4.2020–3.5.2020 (10 days) and 7.5.2020–16.5.2020 (10 days) and one sampling period was immediately after the removal of the lockdown 21.5.2020–30.5.2020 (10 days). We only sampled 10 days after the lockdown

to minimize seasonal changes. We also controlled for the uneven lockdown/no lockdown sampling periods by randomly choosing five sub-samples of 10 lockdown days each, and comparing them to the no-lockdown period. This sampling generated an audio dataset of 388,080 files (30 seconds each). Detailed sampling times for each site are provided in *Supplementary file 1p and q*.

## Automatic acoustic identification and bird activity

Because we only focused on birds (whose call frequencies remain below 10 kHz), to ease file handling, all audio files were first resampled to 22.05 kHz before further analysis (following low-pass-filtering to avoid aliasing). We also high-pass filtered the files with a cutoff at 1 kHz to minimize the interference of low-frequency noise (this was not applied when estimating ambient noise). For each species, we selected a very common syllable (*Figure 1B*) which is typically used for conspecific communication, mostly serving as alarm or attachment calls in *C. corone* and *P. krameri* and as territorial songs in *P. gracilis*. To build an automatic identifier, we first manually selected 138 syllables of *C. corone cornix*, 173 syllables of *P. krameri* and 140 syllables of *P. gracilis* from a worldwide citizen science database of bird recordings (xeno-canto; http://www.xeno-canto.org/). These syllables were used as acoustic templates. We then used spectrogram image cross-correlation to automatically classify sound recordings using Avisoft SASLab Pro 5.1 (R. Specht, Avisoft Bioacoustics, Glienicke, Germany). Spectrogram image cross-correlations is a method for measuring maximum similarity between the spectrograms of the template and the recordings at different time shifts. The output similarity index is a value ranging from 0 to 1. The higher the similarity index, the higher the similarity between template and audio (with 1 meaning that the two spectrograms are identical), and hence the higher the probability that the bird vocalization occurs in the recording. We set the identification threshold to 0.5, and then we manually scrutinized all files with vocalizations above this threshold to make sure that no calls were missed. This high threshold also ensured that we only used high Signal-to-Noise-Ration vocalizations as very weak vocalizations did not correlate with our high-quality templates.

We referred to each file (30 s long) as an 'event'. That is, if we detected a vocalization of one of the three focal species within the file – we considered this as one occurrence of that species. The daily number of events was then used as a proxy for activity in our models (below). Hence, we used three variables to quantify bird activity: (1) The daily number of audio files, which included bird vocalizations of a specific species, was used as a proxy for the daily activity of this species and (2) the daily coefficient of variance (the standard deviation divided by the mean) of the number of audio files estimated in 30 min bins as a proxy of activity variability. (3) We also tested another model in which we used the total number of syllables per file (and not a binary 0/1 value) as the measurement of bird activity.

## Vocalization analysis

Because our results showed that the ambient noise levels during no lockdown were higher than during lockdown (see Results section), we further investigated whether birds modified their vocalizations to mitigate noise interference. We measured two acoustic parameters: (1) To assess changes in calling intensity, we estimated the root mean square (RMS) sound pressure computed over a window defined by a threshold of 20 dB below the peak of the vocalization and (2) To estimate changes in song pitch, we estimated the peak frequency of the vocalizations, that is the frequency with maximum amplitude. Both parameters were estimated from spectrograms computed with the following parameters: Fast Fourier transform with a 512 window; a Hamming window and a 75% overlap between windows, resulting in a temporal resolution of 5.8ms and a frequency resolution: 43 Hz. Vocalizations were analyzed using Avisoft SASLab Pro 5.1.

## Environmental data

We estimated two environmental parameters in the study area: (1) the ambient temperature, (2) the ambient noise level. The environment's temperature was obtained from the Israeli Meteorological service (https://ims.gov.il/en). The temperature was recorded every 10 min. The average temperature of each day was used for the analysis. To estimate ambient noise, we used the average daily sound power as a proxy for ambient noise (at each site). For each bird species, we assessed the noise at the peak-frequency of its vocalization, thus estimating the relevant noise for the species (i.e. Hooded crow: 1600 Hz; Rose-ringed parakeet: 4000 Hz; Graceful prinia: 5000 Hz). We also examined another

noise parameter where the noise was estimated as the power under 1 kHz for all species but this did not alter the results (see **Supplementary file 1r and s**).

## Human activity

To quantify human activity, we ran a random forest classifier implemented using the 'TreeBagger' Matlab function. Firstly, we randomly chose a set of 5960 one second samples from the recording data (taken from all sites). In each site, samples were taken over a 24 hr period from a random day (from both lockdown and no lockdown periods). Second, the content of each of these 5960 samples was identified by listening and identifying the presence of human speech sounds. In total, there were 1294 speech samples and 4666 samples without speech. Third, the 5960 samples were divided into a 'training' set (80% of the samples) and a 'test' set (20% of the samples). The balanced accuracy of the classifier was 75%. Notably, this is far above chance, but more importantly, the accuracy was the same for lockdown and non-lockdown periods so that the comparison between them was fair (and this was our main interest). Finally, we ran the classifier on all recordings and counted the number of speech sounds per 30 s as a proxy for human activity. Note that this method assesses human pedestrian (and not car) activity while the noise measurement (previous section) refers mostly to car noise (**Figure 3— figure supplement 2**).

## Discriminant function analysis

DFA analysis was run in SPSS v20.0 (SPSS Inc, Chicago, IL, U.S.A.) using a leave-one-out-cross-validation procedure. Activity was normalized across sites to a range between 0–1 and we defined 5 levels of activity 0–0.2, 0.2–0.4, 0.4–0.6, 0.6–0.8 and 0.8–1.0. The success of the DFA was determined with a binomial test comparing the DFA's classification performance to a random classification (i.e. chance level of 20%). The prior probabilities of the DFA classes were adjusted to be equal.

## Statistics

To assess the effects of different predictor factors on activity, we run generalized linear mixed models (GLMM) with a Poisson distribution using the function 'glmer' in the R package 'lmerTest' (**Kuznetsova et al., 2017**). We tested the following predictors: the lockdown status (yes/no), ambient temperature, ambient noise, bird species (removed when analyzing each species separately), site category (road, park, residential), human activity and the following interactions: the lockdown status and bird species (removed when analyzing each species separately), the lockdown status and site category, the lockdown status and the count-down (the count-down variable represented the days from the beginning of each lockdown phase to account for temporal dependencies and to represent accumulated effects), the lockdown status and noise. Bird activity was used as the response variable, and the sampling site and sampling time were used as random effects.

In order to check the differences in activity between the no lockdown and lockdown periods for the specific bird-site combinations, we performed Post-hocs using the 'emmeans' function from the 'emmeans' package (**Lenth et al., 2020**). The effect size per parameter (e.g. temperature) was estimated in % by an exponentiation of the coefficient and examining the effect of a change of a single unit in the relevant parameter (e.g., one degree).

To assess the effects of predictor factors on activity variability, we used linear mixed models (LMM), using the function 'lmer' in the R package 'lmerTest' (**Kuznetsova et al., 2017**). The predictors and random variables were the same as for the GLMM. To assure a normal distribution of the residuals prior to fitting the LMM, the activity variability for Graceful prinia was ln transformed while the following parameters were Box-Cox transformed activity variability (of all species together, Hooded crow and Rose-ringed parakeet), the peak frequency and the ambient noise level.

For all models, to select the best model, we used the Akaike information criterion corrected for small sample size (AICc) using the function 'dredge' in the R package 'MuMIn' (**Bartoń, 2015**). The model with the lowest AICc value indicates the best-fitting model. Differences among AIC values were calculated as follows: $\Delta i = AIC_i - AIC_{min}$. Furthermore, $\Delta AICc > 2$ between the first and the second best models is considered as the gold standard for model selection (**Burnham and Anderson, 2002**); therefore, multimodel inference was performed if AIC differences were ≤2, using the model.avg function in the package 'MuMIn' (**Bartoń, 2015**). When several models were fit the data equally (AIC differences of less than 2) we report the average effect size of these models.

We ran several additional models in which the explained parameter was not bird activity. To examine changes in ambient noise between the no lockdown and lockdown periods, a LMM was used with ambient noise as the explained variable, lockdown status and site category as fixed factors, and the specific sites as a random effect. To explore changes in vocal amplitude and peak frequency between the no lockdown and lockdown periods for each bird species, we created a LMM using either vocal amplitude or peak frequency as the explained variable, the lockdown status and noise levels as the independent variables, and sampling site and sampling time as random effects. To assess changes in human activity between the no lockdown and lockdown periods, we ran a GLMM with the lockdown status, the site category and their interactions as explanatory fixed factors, and the sampling site and sampling time as random effects. For all models, the level for statistical significance was set at $\alpha<0.05$. All statistical tests were conducted in R v. 4.1.2 (*Blair, 1996*; *Ross and Browning, 2017*; *Buxton et al., 2018*; *Crooks et al., 2004*; *Estela et al., 2021*; *Gibb et al., 2019*; *Gordo et al., 2021*; *Manenti et al., 2020*; *R Development Core Team, 2021*).

## Acknowledgements

We acknowledge Jiang TL, Feng J, Lucas JR, Li DM for valuable advice and comments on the manuscript. We thank Zhang CM for her assistance with acoustic analysis.

## Additional information

### Funding

| Funder | Grant reference number | Author |
| --- | --- | --- |
| Israeli Ministry of Science | 3-17988 | Yossi Yovel |
| China Scholarship Council | 201906620060 | Congnan Sun |

The funders had no role in study design, data collection and interpretation, or the decision to submit the work for publication.

### Author contributions

Congnan Sun, Data curation, Formal analysis, Funding acquisition, Validation, Investigation, Visualization, Methodology, Writing – original draft; Yoel Hassin, Data curation; Arjan Boonman, Data curation, Writing – review and editing; Assaf Shwartz, Writing – review and editing; Yossi Yovel, Conceptualization, Resources, Supervision, Funding acquisition, Validation, Investigation, Visualization, Methodology, Writing – original draft, Project administration, Writing – review and editing

### Author ORCIDs

Congnan Sun https://orcid.org/0000-0002-9383-6725
Yossi Yovel https://orcid.org/0000-0001-5429-9245

### Ethics

This experiment included passive acoustic recordings of animals in their natural habitat and thus did not require an ethical approval.

Reviewer #1 (Public Review): https://doi.org/10.7554/eLife.88064.3.sa1
Reviewer #2 (Public Review): https://doi.org/10.7554/eLife.88064.3.sa2
Author Response https://doi.org/10.7554/eLife.88064.3.sa3

## Additional files

### Supplementary files

• Source data 1. The bird species, site category, date, bird activity, lockdown type, nose levels, temperature and human activity. The data for *Figure 2A-D* and for *Figure 2—figure supplement 6A-D*.

• Source data 2. The bird species, date, vocal intensity, vocal frequency, lockdown type and nose levels. The data for *Figure 3A-C*, and for *Figure 3—figure supplement 1A-C*.

• Source data 3. The bird species, site category, date, bird activity, lockdown type and human activity. The data for *Figure 2—figure supplement 1A-D*.

• Source data 4. The bird species, site category, date, bird activity, lockdown type and human activity. The data for *Figure 2—figure supplement 2A-D*.

• Source data 5. The bird species, site category, date, bird activity, lockdown type and human activity. The data for *Figure 2—figure supplement 3A-D*.

• Source data 6. The bird species, site category, date, bird activity, lockdown type and human activity. The data for *Figure 2—figure supplement 4A-D*.

• Source data 7. The bird species, site category, date, bird activity, lockdown type and human activity. The data for *Figure 2—figure supplement 5A-D*.

• MDAR checklist

• Supplementary file 1. The sampling time, sampling site, bird activity, statistical results for bird activity and activity variability.

### Data availability

All data generated or analysed during this study are included in the manuscript and supporting file. Source data files have been provided for Figures 1, 2 and 3.

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
