## [Editor Report · eLife assessment]

This manuscript offers a **valuable** contribution to studying wildlife responses during and after COVID-19 lockdowns. It **convincingly** demonstrates that bird species in urban areas respond differently to human activity changes. What sets this study apart from others on avian responses to COVID-19 lockdowns is its use of passive acoustic monitoring. By concurrently measuring anthropogenic noise, a crucial reflection of changes in human activity due to COVID-19 lockdowns, this study reveals rare local-scale variations in bird responses to human activity. Only one study so far has used vocalization recordings to assess the effects of COVID-19 lockdowns on a bird species.

---

## [Referee Report · Reviewer #1 (Public Review)]

Summary:

This study is one of several around the world to investigate how urban wildlife responded to changes in human activity during the lockdowns associated with the COVID-19 pandemic. Unlike several other studies on the topic that used observational data from citizen science programs, this project relied on passive acoustic monitoring to record bird vocalizations during and after stringent lockdown periods in an urban environment. The authors focused on three species that differ in their level of adaptation to human presence, providing an ecologically relevant comparison that highlights the importance of micro-habitats for species living in close proximity to humans.

Strengths:

The element that sets this study apart from most others examining avian responses to COVID-19 lockdowns is the use of passive acoustic monitoring. As the authors describe, this method offers several advantages over other methods (though, it does come with some limitations on what questions can be addressed). Perhaps the most relevant advantage is that it offers the ability to concurrently measure anthropogenic noise in the environment, which is one of the most likely mechanisms for effects on wildlife from changes to human activity. These authors were, therefore, able to show local-scale differences in bird responses to human activity measured at the same scale. To my knowledge, only one other study (Derryberry et al. Science. 2020) has used recordings of vocalizations to examine the influence of COVID-19 lockdowns on a bird species.

It was encouraging to see a study that focused on the local-scale impacts of lockdowns, with methods that could investigate effects within micro-habitats. Logistics prevented many other projects from operating at such fine scales, making the results from this study particularly useful for the examination of rapid changes in bird behavior. This does mean that comparisons between this study and others examining the effects of COVID-19 lockdowns on birds should be done with care, as the effects described here may have been the result of different processes, operating at different spatial and temporal scales. However, that also means this study fills an important gap in our knowledge of how wildlife reacts to human activity in urban spaces.

Weaknesses:

One drawback of the approach is that the acoustic sampling only occurred during the pandemic: samples were taken during several lockdown periods in the early spring (March through early May) of 2020 and then for a period of 10 days after the end of the final lockdown period in late May of 2020. Unfortunately, this means that the interpretation of the effect of lockdowns could have been affected by any shifts in the birds' vocal behavior that resulted from unmodeled environmental factors or normal seasonal phenology during that three-month period. However, the authors chose focal species that would be less prone to seasonal changes in vocal behavior and their approach did account for several factors to minimize any such effects.

---

## [Referee Report · Reviewer #2 (Public Review)]

In this study, the authors tried to gauge the effect of human activity on three species, (1) the Hooded grow, an urban exploiter, (2) the Rose ring parakeet, an invasive, alien species that has adapted to exploit human resources, and (3) the Graceful prinia, an urban adapter, which is relatively shy of humans. A goal of the study was to increase awareness of the importance of urban parks.

Strengths:

Strengths of the study include the fact that it was conducted at 17 different sites, including parks, roads and residential areas, and included three species with different habitat preferences. Each species produced relatively loud and repeatable vocalizations. To avoid the effect of seasonal changes, sounds were sampled within a 10 day period of the lockdown as well as post-lockdown. The analysis included a comparison of the number of sound files, binary values indicating emission of a common syllable, and also the total number of syllables emitted as a measurement of bird activity. Ambient temperatures and sound levels of human activity were also recorded. All of these factors speak to the comprehensive approach and analysis adopted in this study. The results are based on a rigorous statistical analysis, ruling out the effects of various extraneous parameters.

Weaknesses:

Most significant changes may occur near the ambient noise levels and this could lead to a different conclusion, but the authors authors acknowledge this possibility and clarify that they only analyzed vocalizations with high signal-to-noise signals to avoid ambiguity. In the revised version, they also replaced the previous ambient noise parameter with an estimate of ambient noise under 1kHz, assuming that it reflects most anthropogenic noise (not restricted to human speech). This seems reasonable and this new model gave very similar results to the previous one.

In interpreting the data, the authors mention the effect of human activity on bird vocalizations in the context of inter-species predator-prey interactions; however, the presence of humans could also modify intraspecies interactions by acting as triggers for communication of warning and alarm, and/or food calls (as may sometimes be the case) to conspecifics. The behavioral significance of the syllables used to monitor animal activity could be informative in this context; however, the authors acknowledge this possibility in the Discussion. Most importantly, the authors acknowledge the possibility of the above-noted bias, and the potential of a transient nature of the observed effects.

Conclusion:

In general, the authors achieved their aim of illustrating the complexity of the affect of human activity on animal behavior notwithstanding the caveats noted above. Their study also makes it clear that estimating such affects is not simple given the dynamics of animal behavior. For example, seasonality, temperature changes, animal migration and movement, as well as interspecies interactions, such as those related to predator-prey behavior, and inter/intra-species competition in other respects can all play into site-specific changes in the vocal activity of a particular species.

---

## [Author Response]

The following is the authors’ response to the current reviews.

We confirm that that “count-down” parameter, mentioned by reviewer 1, is indeed counted from the first lockdown day and increases continuously, even when we do not have any data – and that this is clearly written in the manuscript.

The following is the authors’ response to the original reviews.

**Reviewer 1:**
(Note, while these authors do reference Derryberry et al., I thought that there could have been much more direct comparison between the results of the two approaches).

We added some more discussion of the differences between the papers.

One important drawback of the approach, which potentially calls into question the authors' conclusions, is that the acoustic sampling only occurred during the pandemic: for several lockdown periods and then for a period of 10 days immediately after the end of the final lockdown period in May of 2020. Several relevant things changed from March to May of 2020, most notably the shift from spring to summer, and the accompanying shift into and through the breeding season (differing for each of the three focal species). Although the statistical methods included an attempt to address this, neither the inclusion of the "count down" variable nor the temperature variable could account for any non-linear effects of breeding phenology on vocal activity. I found the reliance on temperature particularly troubling, because despite the authors' claims that it was "a good proxy of seasonality", an examination of the temperature data revealed a considerable non-linear pattern across much of the study duration. In addition, using a period immediately after the lockdowns as a "no-lockdown" control meant that any lingering or delayed effects of human activity changes in the preceding two months could still have been relevant (not to mention the fact that despite the end of an official lockdown, the pandemic still had dramatic effects on human activity during late May 2020).

In general, the reviewer is correct, and we reformulated some of the text to more carefully address these points. However, we would like to note two things: (1) Changes occurred rapidly with birds rapidly changing their behavior – this is one of the main conclusions of our study, i.e., that urban dwelling animals are highly plastic in behavior. So that lingering effects were unlikely. (2) Changes occurred in both directions, and thus seasonality (which is expected to have a uni-directional effect) cannot explain everything we observed. We are not sure what the reviewer means by ‘considerable non-linear patterns’ when referring to the temperature. Except for ~5 days with temperatures that exceeded the expected average by 3-4 degrees, the temperature increased approximately linearly during the period as expected from seasonality (see Author response image 1). Following the reviewer’s comment, we tested whether exclusion of data from these days changes the results and found no change.

We would like to note that in terms of breeding, all birds were within the same state during both the lockdown and the non-lockdown periods. Parakeets and crows have a long breeding season Feb-end of June with one cycle. They will stay around the nest throughout this season and especially in the peak of the season March-May. Prinias start slightly later at the beginning of March with 2-3 cycles till end of June.

Regarding the comment about human activity, as we now also note in the manuscript, reality in Israel was actually the opposite of the reviewer’s suggestion with people returning to normal behavior towards the end of the lockdown (even before its official removal). We believe that this added noise to our results, and that the effect of the lockdown was probably higher than we observed.

Another weakness of the current version of the manuscript is the use of a supposed "contradiction" in the existing literature to create the context for the present study. Although the various studies cited do have many differences in their results, those other papers lay out many nuanced hypotheses for those differences. Almost none of the studies cited in this manuscript actually reported blanket increases or decreases in urban birds, as suggested here, and each of those papers includes examples of species that showed different responses. To suggest that they are on opposite sides of a supposed dichotomy is a misrepresentation. Many of those other studies also included a larger number of different species, whereas this study focused on three. Finally, this study was completed at a much finer spatial scale than most others and was examining micro-habitat differences rather than patterns apparent across landscapes. I believe that highlighting differences in scale to explain nuanced differences among studies is a much better approach that more accurately adds to the body of literature.

We thank the reviewer for this good feedback and revised the manuscript, accordingly, placing more emphasis on the micro-scale of this study.

Finally a note on L244-247: I would recommend against discounting the possibility that lockdowns resulted in changes to the birds' vocal acoustics, as Derryberry et al. 2020 found, especially while suggesting that their results were the effects of signal processing artifacts. Audio analysis is not my area of expertise, but isn't it possible that the birds did increase call intensity, but were simply not willing (or able) to increase it to the same degree as the additional ambient noise?

This is an important question. The fact is that when ambient noise increases (at the relevant frequency channels), then the measured vocalizations will also increase. There is no way to separate the two effects. Thus, as scientists, when we cannot measure an effect, it is safer not to suggest an effect. Unfortunately, most studies that claim an increase in vocalizations’ intensity in noise, do not account for this potential artifact (and most of them do not estimate noise at a species-specific level as we have done). This has created a lot of “noise” in the field. We do not want to criticize the Derryberry results without analyzing the data, but from reading their methods it does not seem like they took the noise into account in their acoustic measurements. But if you look at their figure 4A you will see a lot of variability in measuring the minimum frequency – which could be strongly affected by ambient noise.

In light of the above, we thus prefer to be careful and not to state changes that are probably false. We added some of this information to the manuscript. We also added the linear equations to the graph (in the caption of figure 3) where it can be seen that the slope is always <= 1.

**Reviewer 2:**
The explanation of methods can be improved. For example, it is not clear if data were low-pass filtered before resampling to avoid aliasing.

We edited the methods and hopefully they are clearer now. Regarding the specific question – yes, an LPF was applied to prevent aliasing before the resampling. This information was added to the manuscript.

It is quite possible that birds move into the trees and further from the recorders with human activity. Since sound level decreases by the square of the distance of the source from the recorders, this could significantly affect the data. As indicated in the Discussion, this is a significant parameter that could not be controlled.

The reviewer is correct, and we addressed this point. Such biases could arise with any type of surveying including manual transects (except for perhaps, placing tags on the animals). We note that we only analyzed high SNR signals and that the species we selected somewhat overcome this bias – both crows and parakeets are not shy and Prinias are anyway shy and prefer to not be out in the open. We would also expect to see a stronger effect for human speech if this was a central phenomenon, and we did not see this, but of course this might have affected our results.

In interpreting the data, the authors mention the effect of human activity on bird vocalizations in the context of inter-species predator-prey interactions; however, the presence of humans could also modify intraspecies interactions by acting as triggers for communication of warning and alarm, and/or food calls (as may sometimes be the case) to conspecifics. Along the same lines, it is important to have a better understanding of the behavioral significance of the syllables used to monitor animal activity in the present study.

We agree with this point and added more discussion of both this potential bias and the type of syllables that were analyzed.

Another potential effect that may influence the results but is difficult to study, relates to the examination of vocalizations near to the ambient noise level. This is the bandwidth of sound levels where most significant changes may occur, for example, due to the Lombard effect demonstrated in bird and bat species. However, as indicated, these are also more difficult to track and quantify. Moreover, human generated noise, other than speech, may be a more relevant factor in influencing acoustic activity of different bird species. Speech, per se, similar to the vocalizations of many other species, may simply enrich the acoustic environment so that the effects observed in the present study may be transient without significant long-term consequences.

We note that we already included a noise parameter (in addition to human speech) in the original manuscript. Following the reviewer’s comment, we examined another factor, namely we replaced the previous ambient noise parameter with an estimate of ambient noise under 1kHz which should reflect most anthropogenic noise (not restricted to human speech). This model gave very similar results to the previous one (which is not very surprising as noise is usually correlated). We added this information to the revised manuscript, and we now also added examples of anthropogenic noise to the supplementary materials (Fig. S8). In general, we accept the comments made by the reviewer, but would like to emphasize that we only analyze high SNR vocalization (and not vocalizations that were close to the noise level). This strategy should have overcome biases that resulted from slight changes in ambient noise.

In general, the authors achieved their aim of illustrating the complexity of the effect of human activity on animal behavior. At the same time, their study also made it clear that estimating such effects is not simple given the dynamics of animal behavior. For example, seasonality, temperature changes, animal migration and movement, as well as interspecies interactions, such as related to predator-prey behavior, and inter/intra-species competition in other respects can all play into site-specific changes in the vocal activity of a particular species.

We completely agree and tried to further emphasize this in the revised manuscript. This is one of the main conclusions of this study – we should be careful when reaching conclusions.

Although the methods used in the present study are statistically rigorous, a multivariate approach and visualization techniques afforded by principal components analysis and multidimensional scaling methods may be more effective in communicating the overall results.

Following this comment, we ran a discriminant function analysis with the parameters of the best model (site category, ambient noise, human activity, temperature and lockdown state) with the task of classifying the level of bird activity. The DFA analysis managed to classify activity significantly above chance and the weights of the parameters revealed some insight about their relative importance. We added this information to the revised manuscript

Suggestions for improvement:In Figure 2, the labeling of the Y-axis in the right panel should be moved to the left, similar to A and C. This will provide clear separation between the two side-to-side panels.

Revised

In Figure 3, it will be good to see the regression lines (as dashed lines) separately for the lockdown and no-lockdown conditions in addition to the overall effect.

Revised

**Editor:**
LimitationsScale: The study's limited spatial and temporal scale was not addressed by the authors, which contrasts with the broader scope of other cited studies. To enhance the significance of the study, acknowledging and clearly highlighting this limitation, along with its potential caveats, modifications in the language used throughout the text would be beneficial. Furthermore, although the authors examined slight variations in habitat, it is important to note that all sites were primarily located within an urban landscape.

We revised the manuscript accordingly.

Control period: The control period is significantly shorter than the lockdown treatment period and occurs at a different time of year, potentially impacting the vocalization patterns of birds due to different annual cycle stages. It is crucial to consider that the control period falls within the pandemic timeframe despite being shortly after the lockdowns ended.

Revised – we included a control comparison to periods of equal length within the lockdown. People gradually stopped obeying the lockdown regulations before its removal so in fact, the official removal date is probably an overestimate for the effect of the lockdown. We now explain this.

RecommendationsHuman-generated noise, beyond speech, might have a greater influence on the acoustic activity of various bird species, but previous studies lacked detailed human activity data. Instead of solely noting the number of human talkers, the authors could quantify other aspects of human activity such as vehicles or overall anthropogenic noise volume. Exploring the relationships between these factors and bird activity at a fine scale, while disentangling them from bird detection, would be compelling. It is important to consider the potential difficulty in resolving other anthropogenic sounds within a specific bandwidth, which could be demonstrated to readers through spectrograms and potential post-pandemic changes. Such information, including daily coefficient of variation/fluctuation rather than absolute frequency spectra, could provide valuable insights.

We note that we have already included an ambient noise factor (in addition to human speech) in the previous version. Following the reviewers’ comments, we examined another factor, namely we replaced the current ambient noise parameter with the ambient noise under 1kHz which should reflect most of anthropogenic noise (not restricted to human speech). This model gave very similar results to the previous one (which is not surprising as noise is usually correlated). We also added several spectrograms in the Supplementary material that show examples of different types of noise.

Authors should limit their data interpretation to the impact of lockdown on behavioral responses within small-scale variations in habitat. A key critique is the assumption that activity changes solely resulted from the lockdown, disregarding other environmental factors and phenology.

Following the editor comment we realized that our conclusion\assertations were not clear. We never claimed that activity changes solely resulted from the lockdown. While revsing the mansucirpt we ensurred that we show a significant effect of temperature, ambient noise and human activity – all of which are not dependent on lockdown. We made an effort to emphasize the complexity of the system. We show that the lockdown seemed to have an additional impact, but we never claimed it was the only factor.

To address this, the authors could compare acoustic monitoring data within a shorter timeframe before and after the lockdown (20 days), while also controlling for temperature effects, to strengthen the validity of their claims. They would need to explain in their discussion, however, that such a comparison may still be confounded by any carry-over effects from the 10 days of treatment.

This analysis would be difficult because although the lockdown was officially removed at a specific date, it was gradually less respected by the citizens and thus the last period of the lockdown was somewhere between lockdown and no-lockdown. This is why we chose the approach of taking 10 days randomly from within the lockdown period and comparing them with the 10 post-lockdown days. We now clarify the reason better.

An option is that authors could frame their analysis as a study of the behavior of wildlife coming out of a lockdown, to draw a distinction from other studies that compared pre-pandemic data to pandemic data.

Good idea – revised.